# Methemoglobinemia Secondary to Inhalation of Automobile Emissions with Suicide Motivations

**DOI:** 10.3390/jcm12030734

**Published:** 2023-01-17

**Authors:** Manuel Antonio Tazón-Varela, Ángel Padilla-Mielgo, Raquel Villaverde-Plazas, Fabiola Espinoza-Cuba, Nekane Gallo-Salazar, Pedro Muñoz-Cacho

**Affiliations:** 1Emergency Department, Laredo Hospital, 39770 Laredo, Spain; 2Community Health Group, Valdecilla Research Institute (IDIVAL), 39011 Santander, Spain; 3Primary Care Unit, Cantabria Health Service, 39011 Santander, Spain

**Keywords:** blood gas analysis, blood gas monitoring, transcutaneous, gas poisoning, methemoglobin, methemoglobinemia, methylene blue, oxygen saturation, oximetry, suicidal ideation, vehicle emissions

## Abstract

Background: Methemoglobinemia (MetHb) is a rare and potentially severe dyshemoglobinemia that can be induced by exposure to oxidizing agents, decreasing the functional capacity of the hemoglobin molecule to transport and release oxygen into the tissues. MetHb can originate from gases with oxidizing capacity generated by internal combustion engines, although since the universalization of catalyst converters in automobiles, a tiny proportion of MetHb poisoning is due to exposure to engine gases and fumes. Within this group, only two cases due to suicidal motivations have been reported in the last 30 years. Case presentation: Here, we expose the case of a patient with MetHb levels of 25.2% (normal 0–1.5%) who with suicidal motivations had attached and locked a hose to the exhaust pipe of her vehicle with electrical tape, becoming exposed to a sustained concentration of the vehicle’s exhaust. Upon her arrival at the emergency department, the presence of generalized greyish cyanosis with alterations of the sensorium, dissociation between saturation measured by arterial blood gas analysis and pulse oximetry (98% vs. 85%), no response to high-flow oxygen therapy, and an excellent response to intravenous methylene blue treatment were highlighted. Conclusions: This report illustrates an original case of acute toxic acquired MetHb due to inhalation of oxidizing substances originating from the bad ignition of an internal combustion engine. When evaluating a patient with suspected gas intoxication, we usually consider poisoning by the most common toxins, such as carbon monoxide or cyanide. In this context, we propose an algorithm to assist in the suspicion of this entity in patients with cyanosis in the emergency department. MetHb poisoning should be suspected, and urgent co-oximetry should be requested when there is no congruence between cyanosis intensity and oxygen saturation measured by pulse oximetry, if there is discordance between the results of oxygen saturation measured by arterial blood gas and pulse oximeter, and if there is no response to oxygen treatment. This algorithm could be useful to not delay diagnosis, improve prognosis, and limit potential sequelae.

## 1. Background

Methemoglobinemia (MetHb) is a dyshemoglobinemia that occurs when oxidizing toxic substances modify the iron of the porphyrin rings of the haem group of hemoglobin, changing them from the ferrous state [Fe^2+^] to the ferric state [Fe^3+^], which interferes with the functional capacity of the hemoglobin molecule to deliver and release oxygen to cell tissues. Methemoglobin does not transport oxygen effectively and shifts the hemoglobin dissociation curve to the left, increasing the affinity for oxygen of the remaining haem groups [1]. Tissue hypoxia occurs without hypoxemia [2]. There are hereditary forms due to deficiency of the reducing enzyme mechanisms and acquired forms due to exposure to oxidizing substances with depletion of the reducing defense systems.

Gases generated by combustion vehicle engines with oxidizing capacity can cause MetHb. However, in the last 30 years, due to mandatory catalytic converters in automobiles, only a few cases of MetHb poisoning secondary to inhalation of exhaust fumes have been reported worldwide [3,4,5,6,7].

The incidence of acquired MetHb is low, and within this range, the incidence of acquired MetHb caused by gas inhalation is lower. In Spain, only 1.36% of intentionally self-inflicted poisonings are due to exposure to gases and vapors [8]. Within this group, the use of gases emitted by car exhausts is a minority and in disuse.

We present the case of a patient with MetHb levels of 25.2% who spent four hours in a car in which engine exhaust conditions had been modified with suicidal motivation.

## 2. Case Presentation

A 41-year-old woman, who is a smoker of 20 cigarettes per day and a cannabis user, was referred to the Emergency Department after being found by a relative in her car (diesel engine, first registered in 2001) with an alteration of the sensorium. She had attached and duct-taped a hose to the vehicle’s exhaust pipe, and the end of the hose was inserted into the passenger compartment. She had concomitantly ingested 7 mg of Lorazepam. When she was rescued after 4 h, the patient told her brother about her suicidal intention.

Upon her arrival at the emergency department, she was hemodynamically stable with tachycardia (115 bpm) and a Glasgow Coma Scale score of 14 points. She had universal greyish cyanosis (Figure 1), headache, and an O_2_ saturation by pulse oximetry (SatO_2_) of 85% breathing oxygen at 15 L/min. Cardiac and respiratory auscultations were normal. In addition to the neurological examination, there was a confused verbal response. A surface electrocardiogram and chest X-ray were normal. The blood collected was chocolate brown in color. Arterial blood gas analysis showed pH at 7.45, arterial pO_2_ at 90 mmHg, arterial pCO_2_ at 32 mmHg, bicarbonate at 22.2 mmol/L, and arterial oxygen saturation (SatpO_2_) at 98%. Blood count levels showed hemoglobin at 14.2 g/dL and 20,300 leukocytes/μL (17,400 neutrophils, 1600 lymphocytes). Coagulation and biochemistry, including ultrasensitive troponin I and C-reactive protein, were normal. The toxicology panel confirmed positivity for benzodiazepines and cannabinoids. The first arterial co-oximetry detected arterial oxyhemoglobin was 70% (normal 94–97%), arterial deoxyhemoglobin was 1.2% (normal 0–0.5%), arterial carboxyhemoglobin was 3.6% (normal 0–1.5%; <10% in smokers), and arterial methemoglobin was 25.2% (normal 0–1.5%).

After treatment with methylene blue (60 mg), neurological normality was restored, her SatO_2_ increased to 90%, and her cutaneous cyanosis decreased. A second co-oximetry 15 min after treatment revealed oxyhemoglobin at 90.6%, deoxyhemoglobin at 0%, carboxyhemoglobin at 3.2%, and methemoglobin at 6.2%. At 45 min, she was asymptomatic, with SatO_2_ at 100%, oxyhemoglobin at 95.4%, deoxyhemoglobin at 0%, carboxyhemoglobin at 2.9%, and methemoglobin at 1.8% in co-oximetry.

The patient was discharged from the Emergency Department after 24 h of observation and later admitted to psychiatry.

## 3. Discussion

Acquired MetHb is a very rare nosological entity that can be induced by toxic exposure to a large number of oxidizing substances. It can be triggered by different situations, from the consumption of poorly preserved vegetable purees to the use of pharmaceutical drugs or drugs of abuse [9,10,11,12] (Table 1). Inhalation of toxic gases is rare. In a study of 138 patients with acquired MetHb, only one patient developed MetHb due to inhalation of toxic gases [13].

One of the difficulties of this pathology is that the clinical picture is chameleonic, and the manifestations depend on the percentage of methemoglobin in the blood and the patient’s baseline characteristics [2]. It can be nonspecific in the early stages, producing anxiety, headache, dizziness, or nausea. At higher levels, tissue hypoxia can lead to lactic acidosis, arrhythmias, convulsions, coma, and death [14]. However, at all times, the common factor is cyanosis with no response to oxygen treatment. Oxygen therapy does not improve oxygen transport to tissues but only produces a hyperoxygenation of the small percentage of oxygen circulating dissolved in the blood and not bound to proteins, which will produce a misleadingly high partial pressure of oxygen and SatpO_2_ in arterial blood gases.

In addition, pulse oximetry can be misleading because methemoglobin interferes with its accuracy. The pulse oximeter allows us to determine the percentage of hemoglobin saturated with oxygen through the emission of infrared light at different wavelengths. Under normal conditions, oxyhemoglobin and deoxyhemoglobin in arterial blood absorb different wavelengths, and from this ratio between absorbances, we obtain the SatO_2_. However, methemoglobin absorbs both wavelengths, although more of the wavelength is normally absorbed by oxyhemoglobin, producing an overestimation of oxyhemoglobin saturation. At methemoglobin percentages <20%, pulse oximetry is normal. When very high percentages are reached, pulse oximetry shows a maintained saturation of approximately 85% regardless of the methemoglobin level. Even if methemoglobin rises to critical levels, the SatO_2_ percentage will not fall below 80% [2]. Thus, the definitive diagnosis will be obtained by co-oximetry, which is urgent in this case.

This gap between Sat pO_2_ in arterial gases and SatO_2_ measured by pulse oximetry is observed both in our case (98% vs. 85%) and in the literature (Table 2).

Regarding treatment, high-flow oxygen therapy at increasing doses is not effective. The cornerstone of urgent treatment, in addition to stopping exposure to the toxicant by evacuating the patient from the contaminated area, is methylene blue (MB), which would be indicated in symptomatic patients or with MetHb levels above 20%. This antidote’s action is to return methemoglobin to oxyhemoglobin through two mechanisms. MB accepts electrons for NADPH-methemoglobin reductase, increasing its reductive activity. In addition, MB is reduced to leucomethylene blue with the ability to directly reduce methemoglobin [14]. This will occur as long as there is no glucose-6-phosphate dehydrogenase deficiency. If we treat a patient with a glucose-6-phosphate dehydrogenase deficiency with MB, we may increase the MetHb.

MB would also not be indicated in patients being treated with selective serotonin reuptake inhibitors, as MB inhibits monoamine oxidase A, which may increase intersynaptic serotonin levels, with the risk of triggering serotonin syndrome.

For the treatment of MetHb, an initial dose of 1–2 mg/kg is recommended and may be repeated if necessary. When MB is ineffective, urgent exanguinotransfusion is an option [1].

Ascorbic acid, as an oxidative stress reducer, is not usually applied in ED due to its slow action. Other treatments, such as N-acetylcysteine, hemodialysis, or hyperbaric chamber, are subject to debate [7].

How do we link engine exhaust inhalation and MetHb? Diesel combustion produces toxic gases such as nitrogen oxides, carbon dioxide, sulfur dioxide, formaldehyde, or benzene, which can induce MetHb [3,4,5,6]. Even the ingestion of naphthalene pellets, a natural component of diesel, produces MetHb [15].

In 1993, the European Union obliged producers of internal combustion vehicles to include a catalytic converter in their models. It is a honeycomb-shaped ceramic device whose cells are impregnated with a resin containing platinum, palladium, and rhodium, which produces a molecular rearrangement that transforms the nitrogen oxide, carbon monoxide, and hydrocarbons contained in the exhaust gases into harmless nitrogen, oxygen, and water. In our case, this device should have prevented the production of oxidizing substances. Since the patient had no medical history that would lead us to assume hereditary MetHb, we conjecture that although the car had passed the mandatory vehicle inspection a year earlier, it is likely that due to its age, the catalytic converter was not working properly, and the gases that passed into the passenger compartment of the vehicle oxidized the hemoglobin.

This report illustrates an original case of acute acquired methemoglobinemia due to toxic inhalation of oxidizing substances from the bad ignition of an internal combustion engine, which to our knowledge is the first published case in Europe of a surviving patient with methemoglobin poisoning secondary to inhalation of engine fumes with suicidal intention.

As this condition is rarely seen in the emergency department, it is not usually included in the initial differential diagnosis of emergency physicians when approaching patients with cyanosis. Therefore, we propose an algorithm to aid in the diagnostic suspicion of MetHb in patients with cyanosis in the ED (Table 3). We suspect methemoglobinemia intoxication and request urgent co-oximetry when there is no congruence between the intensity of cyanosis and oxygen saturation measured by pulse oximetry, there is discordance between the results of oxygen saturation measured by arterial blood gases and pulse oximetry, and the cyanosis is not corrected by oxygen supplementation. This model to aid in the suspicion of MetHb could be useful to avoid delaying diagnosis in these patients, improving prognosis, and limiting potential sequelae. Therefore, as soon as exposure to a methemoglobin generator is suspected, CO-oximetry is absolutely imperative.

## Figures and Tables

**Figure 1 jcm-12-00734-f001:**
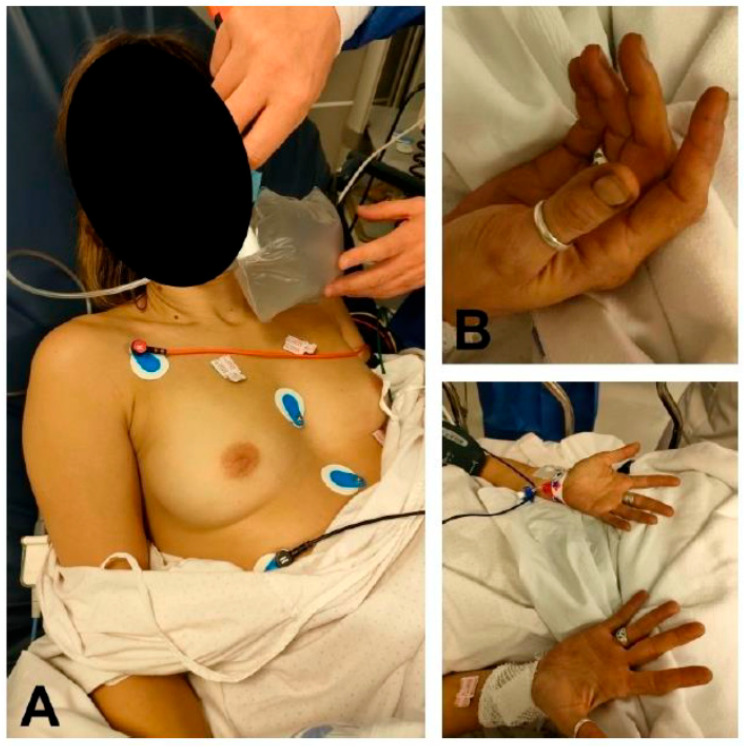
Patient with methemoglobinemia secondary to inhalation of gases from a vehicle engine. (**A**) Cyanotic discoloration of the patient’s skin compared to the skin on the back of the physician’s hand. (**B**) Peripheral cyanosis. Note the grayish discoloration of the patient’s nail beds.

**Table 1 jcm-12-00734-t001:** Causes of acquired methemoglobinemia.

**Pharmaceutical drugs**	Antibiotics	Dapsone	**Food poisoning**	Foods rich in nitrates	Spinach
		Trimethoprim/Sulfamethoxazole			Chard
	Local anesthetic	Benzocaine			Broad beans
		Bupivacaine			Aurugula
		Lidocaine			Lettuce
		Prilocaine			Watercress
	Antineoplastic agents	Flutamide			Beans
		Ciclofosfamide			Broccoli
	Nitrites-Nitrates	Nitroglycerine			Carrots
		Nitroprusside			Choy Sum
		Silver nitrate		Water contaminated with nitrite-producing bacteria	
		Amyl nitrite		Food additives	Sodium nitrite
	Antimalarials	Primaquine			Potassium ferrocyanide
		Cloroquine		Spices	Asafoetida splint
	Antiemétics	Metoclopramide			
	Hypnotics	Zopiclone		Nitrates present in frozen foods	
	Analgesics	Phenazopyridine			
	Hypouricemic	Rasburicase			
**Cosmetics and household products**	Hair dyes	Hydrogen peroxide	**Diseases and other medical conditions**	Hemodialysis water contaminated with hydrogen peroxide	
		Anilines			
		Paraphenylenediamine		Use of nitric oxide as pulmonary vasodilator	
	Synthetic dyes for shoes and clothing	Anilines			
		Acetanilide		Infection by exposure to nitrite-reducing bacteria	
		Azole dyes			
		Sodium nitrite		Electrosurgical scalpel smoke	Benzene
	Stain remover	Anilines			Hydrogen cyanide
	Mothball	Naphthalene			Formaldehyde
	Solvents	Benzene derivates			Acrylonitrile
		Dimethylaminobenzene	**Industrial components**	Metal industry	Taladrine
**Gas inhalation**	Combustion engine exhaust gases	Nitric oxide			
		Carbon dioxide		Paper manufacturing	Sodium chlorite
		Formaldehyde			
		Benzene		Precious metal mining	Potassium cyanide
	Electronic cigarette	Vanillin			
		Propilenglicol		Engine oil	Nitrobenzene
**Environmental pollutants**	Pesticides	Iodates			
		Chlorates		Engine antifreeze	Monocarboxylic acid
		Bromates			
		Dinitrophenol	**Recreational drugs**	Amyl nitrile	
		Propanil			
	Herbicides	Paraquat			
		Copper sulfate			
	Insecticides	Aluminum phosphide		Butyl nitrite (poppers)	
		Metaflumizone			
		Indoxacarb			

**Table 2 jcm-12-00734-t002:** Pulse oximetry, blood gases, and co-oximetry in Methemoglobinemia.

Suspected Methemoglobinemia in a Patient with Cyanosis						
Reference	SatO_2_ (%)	SatpO_2_ Pre (%)	SatpO_2_ Post (%)	PaO_2_ (mmHg)	CarboxiHb (%)	MetHb (%)
Laney et al. [3]	-	70	95	118	0	24.8
Suyama et al. [4]	-	-	86	236.2	0	44.3
Kumagai et al. [5]	82	-	98	348	32.6	24.2
Vevelstad et al. [6] †	-	-	-	-	0.6	56.3
Cho et al. [7]	88	-	93	201.8	0	59.6
Tazón-Varela et al.	85	-	98	90	3.6	25.2

CarboxyHb: carboxyhemoglobin; MetHb: methemoglobin; PaO_2_: partial pressure of oxygen in arterial blood; SatO_2_: oxygen saturation measured by pulse oximetry; SatpO_2_: oxygen saturation measured by arterial blood gas analysis; SatpO_2_ post: oxygen saturation measured by arterial blood gas after oxygen administration; SatpO_2_ pre: oxygen saturation measured by arterial blood gas without prior oxygen administration. † Values in cadaver for completed suicide.

**Table 3 jcm-12-00734-t003:** Methemoglobinemia. Diagnostic algorithm in the emergency room

Suspected Methemoglobinemia in a Patient with Cyanosis.
Investigate toxic exposure
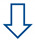
Assess congruence between SatO_2_ and cyano intensity ^a^
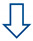
Assess congruence between SatO_2_ measured by pulse oximetry and SatpO_2_ measured by arterial blood gas ^b^
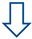
Assess the response to treatment with O_2_ ^c^
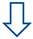
Request urgent CO-oximetry ^d^

^a^ Obvious cyanosis with normal or slightly altered pulse oximetry. ^b^ Low saturation by pulse oximetry and normal saturation by arterial blood gas. ^c^ Does not improve with high-flow oxygen therapy. ^d^ Diagnosis.

## Data Availability

The datasets used and/or analyzed during the current study are available from the corresponding author on reasonable request.

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
