# Peer review of "Methemoglobinemia Secondary to Inhalation of Automobile Emissions with Suicide Motivations"

_jcm, 2023, doi:10.3390/jcm12030734_

Round 1
Reviewer 1 Report
This case report on an unfortunate individual who was moved to try to end their lives through the use of their vehicle emissions is of interest from several standpoints. Firstly, it is now popularly believed that it is probably not possible to cause serious injury to oneself from automobile emissions and this study provides a relevant repost to this view.
Secondly, the perennial issue of the failure of pulse oximetry to detect methaemoglobin toxicity is of course not a new problem (Coleman and Coleman, Drug induced methaemoglobinaemia, Drug Safety 14, 394–405 1996) however, it is very important clinically to keep drawing this issue to Emergency Room personnel's attention.
Thirdly, it is of interest, (and the authors draw attention to this) that a very wide range of ingested/inhaled and parentally administered agents can cause methaemoglobin formation and they do not necessarily have to be aromatic amines.
The authors propose that the reason oxidant agents were present in the emissions were probably due to aging of the catalytic convertor, which is fair comment. However, depending on the fuel that was used, it is also likely that the conditions (extended idling over many hours) promoted the formation of unburned multi-ring hydrocarbons and nitrogen oxides (Roy et al., Cold start idle emissions from a modern Tier-4 turbo-charged diesel engine fueled with diesel-biodiesel, diesel-biodiesel-ethanol,and diesel-biodiesel-diethyl ether blendsApplied Energy 180 (2016) 52–65).
Minor issues...
Abstract: 'modifying the exhaust conditions of the engine.' I see what the authors might mean from this, but where the exhaust is directed has no bearing on the combustion process so does not actually impact the gaseous and particulate composition of the emissions. So perhaps modify to the patient becoming exposed to a sustained concentration of the vehicle's exhaust, or something like that.
'her autolytic motivation' again I see what the authors mean, but perhaps change for 'her suicidal intention'
The diagnostic algorithm is useful, but from a practical perspective, as soon as exposure to a methaemoglobin generator is suspected, CO-oximetry is absolutely imperative.
one was cited as 15 but was actually listed as 16
Author Response
We agree with all the proposed modifications.
- The text in the abstract has been changed to: “becoming exposed to a sustained concentration of the vehicle's exhaust”.
- Has been replaced: “autolytic motivation” by “suicidal intention”.
- A sentence has been added to the text, following the reviewer's comment regarding the algorithm.
- The error in the numbering of the bibliography has been corrected.
Reviewer 2 Report
A case of intoxication with exhaust gas from a old car, which is a rather rare condition since cars are equipped with catalytic converters. The patient is a smoker and cannabis user. Were these elements the source of confusion during admission to the emergency room?
Author Response
We appreciate the observation, although we believe that in this case the consumption of tobacco and cannabis was not a reason for confusion during admission to the emergency room.